# The Mediating Role of Customer Satisfaction between Antecedent Factors and Brand Loyalty for the Shopee Application

**DOI:** 10.3390/bs13070563

**Published:** 2023-07-06

**Authors:** Solomon Gbene Zaato, Noor Raihani Zainol, Sania Khan, Ateekh Ur Rehman, Mohammad Rishad Faridi, Ali Ahmed Khan

**Affiliations:** 1Faculty of Entrepreneurship and Business, Universiti Malaysia Kelantan, Kota Bharu 16100, Malaysia; solomon.gz@umk.edu.my (S.G.Z.); raihani@umk.edu.my (N.R.Z.); 2Department of Human Resource Management, College of Business Administration, Prince Sattam Bin Abdulaziz University, Al Kharj 11942, Saudi Arabia; 3Department of Industrial Engineering, College of Engineering, King Saud University, Riyadh 11421, Saudi Arabia; arehman@ksu.edu.sa; 4Department of Marketing, College of Business Administration, Prince Sattam Bin Abdulaziz University, Al Kharj 11942, Saudi Arabia; m.faridi@psau.edu.sa; 5Department of Management Information Systems, College of Business Administration, Prince Sattam Bin Abdulaziz University, Al Kharj 11942, Saudi Arabia; a.gulam@psau.edu.sa

**Keywords:** accuracy and price of delivery, information quality, ease of payment, Shopee application, security of payment, customer satisfaction, brand loyalty

## Abstract

Lately, smartphones have had a significant impact on how individuals act, mainly when they shop. In Malaysia, Shopee is the online shopping website that has garnered the most traffic from e-commerce sites. Shopee Express recognizes the importance of customer happiness and brand loyalty in measuring consumer purchasing behavior for long-term performance. Some prior studies have had mixed results on the factors that influence online shopping apps’ brand loyalty with the interactive effect of other variables. To contribute to resolving these varied views, this study proposes that customer satisfaction mediates the antecedent factors that influence students’ loyalty to the Shopee app. As a cross-sectional online survey, we obtained data from 298 university students using the Shopee application. Partial least squares structural equation modeling (PLS-SEM) was used to analyze data, whereby the results confirmed a significant effect of the accuracy of delivery on brand loyalty and customer satisfaction, the price of delivery and customer satisfaction, and information quality on brand loyalty and customer satisfaction. Furthermore, easy payment also significantly affected brand loyalty and customer satisfaction, and customer satisfaction mediates delivery accuracy and brand loyalty, the price of delivery, information quality, and the brand loyalty link of students to the Shopee application. This study’s novelty is uncovering the intervening role of customer satisfaction in the antecedent factors of brand loyalty of the Shopee app. This study further contributes by providing helpful information to the stream of online logistic firms like Shopee to meet client needs and by providing valuable insights for scholars.

## 1. Introduction

Smartphones have contributed to changing people’s behavior and easy interaction, with incessant interest in online shopping using various mobile applications worldwide [1]. The rapid rise in online spending has pushed logistic service providers (LSP) to play an essential role in facilitating the movement of goods, which has also increased consumer demand for specialized delivery services and increased the use of couriers, with online shoppers expecting high standards for the online delivery of their packages [2]. Today, logistics firms provide a real-time shopping platform application, also known as Shopee, as an alternative to meet the needs of consumers for quick response. 

In Asia, competition in the e-commerce industry is still dominated by large companies like Shopee. Shopee was first started as a customer-to-customer (C2C) marketplace but shifted to business-to-customer (B2C) [3]. Research shows several factors that can influence changes in consumer behavior, including consumer experience, ease of access to products, competitive pricing, product quality, and several other factors that can influence these changes [4]. The use of logistics commerce applications is now one of the trending phenomena in which people prefer to buy daily necessities online. Many courier companies have been established locally and abroad to deliver goods to customers, such as J&T Express, GD Express, City-Link Express, DHL, and Shopee Express in Malaysia. In [5], Malaysia’s Ministry of Domestic Trade and Consumer Affairs (KPDNHEP) acknowledged 186 complaints about Shopee Express operations from January to 30 June 2021. Through their complaints, consumers sought compensation totaling MYR 9700.92. Customers were not provided with their requested goods in 94 incidents, and the most frequent complaints were about missing deliveries (33 instances), defective products (14 incidents), and goods packed differently than intended (goods). This indicates that the Shopee app and other e-commerce operators need to be mindful of their client’s satisfaction issues and restore trust among them [6]. Logistics firms can use the Shopee website to build customer trust and encourage them to shop. Continuous earnings and revenue are generated by devoted consumers, with the difficulty of obtaining prospective clients decreasing [7]. To attract repeat customers, logistics companies must keep them satisfied by offering attractive services and offers, which can be achieved by providing high-quality services. Service quality is believed to lead to customer satisfaction, loyalty, or repeated business [8,9].

The most common problems during the shipping process are human error during the receiving process, no automated system for picking, bad route planning, and driver error. Next, the dumping of delivery services in Malaysia has raised issues when many companies charge high customer fees. Accordingly, there are no guidelines from the government to control and set maximum charges for delivery services, with shipping agencies offering aggressive charges as a choice for customers [9]. To fully tackle the COVID-19 crisis, efficient delivery processes and payment systems for goods and services and customer satisfaction are therefore essential. As a result, firms require good strategies to continuously evaluate and improve their different service activities, such as from addressing customer queries and complaints to solving customers’ problems and meeting their expectations [9]. From [10], brand loyalty determinants were classified as free sampling, perceived brand value, training, and word-of-mouth communication to keep customers from switching to other brands as variables contributing to brand loyalty. Similarly, [11], a study on airline service delivery, was based on airline tangibility, terminal tangibility, and affinity with an airline’s image as the basis for enhancing the level of perceived service quality in passengers’ satisfaction. 

Other prior studies, like [12], detected the need for firms to extend the level of client satisfaction to sustain the level of client loyalty. In [13], researchers attempted to identify factors influencing customer satisfaction from various perspectives and contexts since people have different perceptions of these factors with brand loyalty. Furthermore, customers can indicate whether a particular product or service exceeds or falls below their expectations and whether they consider a specific service essential. Most studies focus on the accuracy of delivery orders, prices, and information quality with less incorporation of ease of payment and security of payment explained by client satisfaction and brand loyalty as per [12,14]. In [15], researchers emphasized the role of digital transformation, which impacts mobile applications, accelerating customer-centric experiences, dramatically improving velocity, capturing value, etc., which enables agility and competitiveness.

Nowadays, it is common to see that many young people have a penchant for shopping using Shopee applications, whereas merchants prefer to use Shopee Express to transfer goods to their customers. However, few studies exist on Shopee applications [1,16], particularly in Kelantan, North East state of Malaysia. Interestingly, most undergraduate students in the state of Kelantan most often use the internet to shop and use Shopee Express to send and receive goods. However, fewer studies examined antecedent variables such as delivery accuracy, pricing, information quality, payment, speed, and security. Whereas, their level of happiness and loyalty to the Shopee application are explained by other variables like customer satisfaction on brand loyalty to online delivery services like the Shopee app of undergraduate students of Kelantan. In view of the extant studies, this novel study which reflects the ServQual model deemed suitable to fill the research gap. The aim of this study, along with the rest of the research described, is to examine the mediating effect of customer satisfaction between antecedent factors and brand loyalty to the Shopee app. 

## 2. Literature and Hypotheses Development

### 2.1. ServQual Theory

The electronic service quality (e-SERVQUAL) model [17] serves as the theoretical basis of this study. It is used to access how customers judge e-service quality to measure the service quality of any applications like the Shopee app [3]. Service quality (ServQual) features a three-phase method that includes multiple stages of collecting and analyzing empirical data as well as informal interviews. Effectiveness, satisfaction, fulfilment, confidentiality, accessibility, reimbursement, and interaction are its main divisions.

Concerning the ServQual model, the information quality of Shopee applications best provides a visible quality dimension. It covers overall design, ease of payment and use. Thus, Shopee applications’ information quality can be evaluated from various perspectives, such as product quality, system quality, software or service quality. Based on the literature, a good relationship between service providers and customers can exist when appropriate information and services meet customer needs. This is essential for marketing products or services and providing information that can attract customers to buy [18,19]. Shopee pay reliability is the next ServQual dimension of Shopee applications service, a service-based capability that provides customers with confidence and firm resolution. Customers expect search engines and payment facilities, and they also expect information to be presented reliably in offline services. The correct technical functionality of the website and or technical aspects of the user interface drive the perception of reliability. In contrast, the resultant aspects are evident in the accuracy of services promised, billing, and product information [20]. This is the backbone of Shopee applications service quality. With a convenient Shopee applications checkout service, customers find another site to use the checkout feature, where customers might get surprised and may stop and conclude that what is being done could be more effective and efficient [21].

In the ServQual model, the accuracy of delivery is evident from the quality dimension of responsiveness by online service providers. Service providers’ responsiveness to customer needs, whether timely or not, is captured by the responsiveness and fulfilment dimensions [22]. Thus, Shopee delivers its products to customers quickly, efficiently and at the right time with accurate billing processes, being instantly connected, and having full access to logistics information. The accuracy of Shopee application content delivery orders can be related to the reliability of logistic quality, making it simpler for customers to perceive lower risk, provide a stronger rationale for their choice and make quick decisions. These factors influence customer loyalty and willingness to purchase online [23]. Thus, customer loyalty and service rating are influenced by how quickly a service provider responds to customer requests. Thereby, assurance is a crucial quality component that customer service agents utilize to gain the trust of their clients [24]. In terms of the Shopee app, the shipping cost best offers the quality dimension of assurance. Consumer responses to pricing are influenced by knowledge and details regarding product or transaction prices, as well as accessibility [25]. Price continues to be a key factor motivating users to create Shopee applications. As a result, the price remains a vital driver influencing consumers to make Shopee applications. It is also observed that pricing is an effective way to motivate price-sensitive consumers to get greater value or to buy products at lower prices, which leads to customer satisfaction and brand loyalty [25].

Protecting personal and financial information is referred to as privacy or security, as is the degree to which users believe the website is secure from intrusion [26], and for users of online banking, security is thought to be a crucial component in terms of judging the caliber of electronic services when accessing the website and making purchases [27]. Shopee customers usually need access to employees or the company’s physical facilities, ensuring that trust is established in different ways. A reliable online payment solution can increase customer trust. In addition, security deals with maintaining security and privacy dimensions during the online transaction process, including the personal data entered by the customer on the Shopee application. A lack of security generates mistrust among customers, which lowers satisfaction and loyalty [12]. Thus, the most important aspect of Shopee applications’ service quality is payment security. Therefore, the application of the ServQual model in this study justifies the choice of variables and fills the research gap.

### 2.2. Effect of Accuracy of Delivery Order (ACU) on Brand Loyalty (BRL)

According to the pertinent literature, a sign of a company’s overall strength is how timely or precise their deliveries are. Any product’s delivery service connects to the supply chain but also works directly with and for the customers. Customers who are pleased with the delivery service will be loyal customers and will place additional orders. [28,29]. Therefore, delivery accuracy is essential for online buyers. So, in accordance with [29], the delivery date and product quality will have an impact on consumer satisfaction and loyalty. We consequently proposed the following hypotheses:

**H1a.** 
*ACU positively and significantly affects BRL and CSAT to the Shopee app.*


**H1b.** 
*CSAT positively mediates the link between ACU and BRL and the Shopee app.*


### 2.3. Effect of Price of Delivery (PRD) on Brand Loyalty (BRL)

Price is a major factor that might influence a customer’s decision to buy and has a significant impact on that decision [30,31]. Shopee apps developed Shopee Express after being the most widely used online shopping app in Malaysia. In the past, Shoppe Corporate has transported goods via outside logistic firms like J&T and POSLaju. It is evident that [32] price is a major factor that might influence a customer’s decision to buy and has a significant impact on that decision through consumer-based brand equity and product innovation. Similarly, a study [33] on buying and deciding to buy shows that pricing significantly affects these variables in addition to customer satisfaction and brand loyalty. The following theoretical views are put forth because Shoppe Express provides the lowest service pricing compared to third-party logistics firms, making Shopee brand apps a viable option that encourages devotion to the brand based on consumer happiness.

**H2a.** 
*PRD positively and significantly affects the BRL and CSAT of the Shopee app.*


**H2b.** 
*CSAT positively mediates the link between PRD and BRL and the Shopee app.*


### 2.4. Effect of Information Quality (INQ) on Brand Loyalty (BRL)

Information quality on logistics businesses has been widely utilized in this business sector. In the case of the official Shopee brand website, INQ notifies clients about their purchase and allows them to open the app directly or browse it using the ordering page [34]. Shopee developed the concept of Shopee Express to lessen the financial burden placed on consumers having third-party logistics firms. The efforts made by Shopee to provide courier services are made for products that are separate from its commitment to enhancing the caliber of delivery providers and winning the trust and loyalty of its customers [35]. According to researchers [36], they examined the impact of information quality, customer experience, pricing, and service quality on purchase intention with customer perceived value as a mediator variable. Therefore, we suggested the following hypostheses:

**H3a.** 
*INQ positively and significantly affects the BRL and CSAT of the Shopee app.*


**H3b.** 
*CSAT positively mediates the link between INQ and BRL and the Shopee app.*


### 2.5. Effect of Ease of Payment (EPA) on Brand Loyalty (BRL)

The payment methods used by online service providers influence the user’s decision to purchase since they increase brand loyalty and customer satisfaction [37]. Shopee app provides a variety of payment options, including cash on delivery, internet transfers, Shopee pay wallet, Shopee pay later, cash payments at grocery stores, payment plans, and so on. The enormous consumer base of Shopee consists of internal and external elements. Customers from both urban and rural locations, who make up a large portion of Shopee’s consumer base, can opt for cash after delivery (COD) as payment. Students using the Shopee application as their online buying gateway found COD more straightforward [38]. These services are not provided to users of all online apps or service providers. Customers who experience payment difficulties, particularly with online banking or CDM due to demographic considerations, will stick with the Shopee brand apps and be happy with their service. Studies [38,39] show that when utilizing an online application that makes payments simple, customers will be content, and brand loyalty will be raised as a result. Therefore, we proposed that:

**H4a.** 
*EPA has a positive and significant effect on BRL and CSAT on the Shopee app.*


**H4b.** 
*CSAT positively mediates the link between EPA and BRL and the Shopee app.*


### 2.6. Effect of Payment Security (SPA) on Brand Loyalty (BRL)

When it comes to online purchasing platforms, the security of payments is seen as a crucial component of customer fulfilment and loyalty [38]. As customers shop online and provide sensitive information, such as their names and financial information, during the transaction, if a hacker obtains this information, it could be dangerous, and if Shopee security is unable to protect its users, it could be easily made public online [40]. It is also reported that [16] over fifty percent of Malaysian internet shoppers are concerned about their privacy and security when making purchases. These concerns comprise the fraudulent use of credit cards, fictitious online sellers, financial data, theft of personal data, and an overall absence of confidence, whereas other researchers [41] reported that customers who are happy will buy from an online platform that is secure and has an excellent record for handling customers’ personal data and confidentiality, such as Shopee. Safety constitutes a crucial component that influences consumer interest in e-commerce.

Additionally, studies by researchers like [42,43] reported that client loyalty, the caliber of delivery, and consumer happiness are pertinent to e-commerce business through online shops. Likewise, it is evident that customer service quality substantially impacts brand loyalty, repeat business, and intentions to repurchase [44]. Thus, when clients are satisfied with the services, they have more motivation to make subsequent purchases or stick with the company because of the accuracy of delivery, payment security, and cost of delivery that Shopee provides its users [45]. Based on the above argument, it was proposed that.

**H5a.** 
*SPA positively and significantly affects BRL and CSAT to the Shopee app.*


**H5b.** 
*CSAT positively mediates the link between SPA and BRL and the Shopee app.*


### 2.7. Customer Satisfaction (CSAT)

Customer satisfaction measures customers’ happiness with a company’s goods, services, and abilities. The Latin words “Satis” (thus, sufficient) and “end faction” (Latin, facere—to do or to be) are the source of the term satisfaction [46]. Customer satisfaction, or CSAT, with a company’s goods or services occurs when certain behavioral requirements, including accurate delivery processes, affordable delivery, accurate information, simple ways of paying, and secure payments methods, are met. A person’s general attitude towards a product after acquiring it is referred to as their “satisfied or dissatisfied consumer” behavior. Based on their prior psychological, behavioral, and mental situations, each client anticipates superior products or service prospects, such as specifications, benefits, or details, to provide unique values [47,48].

Based on Oliver’s theory, the overall satisfaction approach sees customer satisfaction as a cumulative assessment that includes the sum of particular and varied characteristics of satisfaction related to the product. Customer impressions of a company’s performance in terms of its products and services are captured by overall satisfaction instead of specific transactional satisfaction. Customer happiness is crucial since it allows us to gauge the major impact on businesses’ long-term profitability and consumers’ purchasing patterns. Lower prices, next-day delivery, and, most of the time, payments are completed when the goods are delivered are some of Shopee’s selling points [5]. Customers may become upset whenever their personal information is utilized against their permission. In addition to being annoyed by spam and newsletter emails and worried about security when transmitting credit card information online, customers may also have quality concerns if a website fails to deliver the information they seek. Customers’ satisfaction and brand loyalty may be impacted by the quality of the information provided and the simplicity of payment. Customer satisfaction subset variables such as customer loyalty, love, and passion towards brands have been widely researched in the past. A breakthrough in understanding the phenomenon of brand fidelity spurs a novel perspective in the journey of customer satisfaction [49].

Additionally, subpar goods and delayed deliveries impact consumer satisfaction and brand loyalty [50]. Reflecting on customers’ evaluations of a firm’s product or service in terms of accuracy of delivery, price of delivery, information quality, ease of payment and security of payments, study [43] shows that customer satisfaction has a positive and significant effect on brand loyalty. This causes an increase in satisfied customers, leading to brand loyalty and repetitive purchase of the same brand and product. Customer satisfaction is crucial for preserving customer loyalty over time and greatly impacts brand loyalty [51,52]. They are satisfied as their expectations are met, and their experiences align with the company’s products. As a result, the hypothesis that follows is put forth:

**H6.** 
*CSAT significantly influences the BRL to the Shopee Application.*


### 2.8. Brand Loyalty (BRL) of Undergraduate Students to Shopee Application

Brand loyalty (BRL) is the conduct of customers who exhibit their confidence in a particular brand by continually consuming or utilizing products from that brand instead of rival brands. Indicators of brand loyalty include favorable referrals, customer fulfilment, brand trust, and fair prices [53]. When a customer consistently buys a product made by the same business rather than an alternative product made by a rival, such a customer is referred to as a “brand loyalist.” Brand loyalty is an important concept in businesses since it increases profitability and spurs businesses to attract new clients. According to a prior study by [54], consumer trust and fulfilment with the product or service are typically the foundations of loyalty. Client fulfilment and loyalty to the brand are also affected by price, quality of information, delivery accuracy, and payments. As reported, customer satisfaction and BRL to applications such as Shopee are based on customers’ assessment of ACU services and delivery on time, which are considered essential satisfaction factors [55].

CSAT and brand loyalty are essential for a company to succeed. Businesses that operate online typically take steps to meet the needs of customers as quickly as possible and continue to work to increase consumer online buying habits by offering high-quality goods and services [56]. Regarding the Shopee app, promptness is related to delivery accuracy and the system of information on the app that guarantees customer happiness and repeats business. Satisfied customers are likelier to make additional purchases and stick with a brand like the Shopee app. After purchases, customers’ ratings and reviews provide feedback encouraging new customers to try the goods or services that online service providers like Shopee offer [30,55].

## 3. Methodology Adopted

To estimate the effect of customer satisfaction and the mediating factors that influence brand loyalty, a multi-item measurement scale is adopted. The details of it are presented here below.

### 3.1. Sample Size Determination and Data Collection Procedure

This study examined the effect of customer satisfaction mediating the factors influencing brand loyalty to Shopee of undergraduate and postgraduate students from Kelantan in the northeastern Peninsular state of Malaysia as its sampling frame. A cross-sectional and quantitative study was espoused, and based on the number of study constructs and population, G-power computation provided a nominal sample size of 187. As per [57], 10 or up to 20 percent of the main sample size can be pilot-tested; hence, we pre-tested 40 people where the questions were shared via WhatsApp group, as shown in Table 1, to test the content and the reliability validities of the study variables. Preliminary analysis from the SPSS proved Cronbach’s alpha values of over 0.70 achieving a suitable reliability test. The pre-test results provided good justifications for the study before the questionnaire distribution. The final questionnaire was designed via a Google Forms and distributed via social media for easy data collection and analysis from the respondents. In sum, we retrieved 312 respondents and yielded 298 considered appropriate through Google form online survey delivery in March 2023 after extracting the incomplete questionnaires. First, the questionnaire stated the study’s rationale and obtained respondents’ consent to participate in the study.

### 3.2. Development of Measurement Scales

The study used multi-item scales to measure constructs as follows: Accuracy of Delivery Order, Price of Delivery, Information Quality, Ease of Payment, and Payment Security (Independent variables), Customer Satisfaction (Mediating variable) and Brand Loyalty (Dependent variable). In designing these items, we first considered how to reduce bias likely to occur in a single-source cross-sectional study. In order to achieve, if possible, fair and reliable responses, we created a questionnaire based on previously established procedures employing several Likert scales. In order to determine whether subjects were in disagreement or agreement with the questionnaire items regarding the independent and mediating variables, a five-point (5) Likert scale was utilized. Then, a scoring system (1 = strongly disagree to 5 = strongly agree) was applied. A seven-point (7) Likert scale was employed for the dependent variable, with 1 denoting strong disagreement and 7 denoting strong agreement. For the questionnaire items, five items were changed from each of the related research. Refer to Appendix A in regard to the accuracy of delivery [58,59], price of delivery [60,61], information quality [45,62,63], ease of payment [58,59,64], payment security [65,66] and customer satisfaction [36,67] with brand loyalty items [68,69,70].

## 4. Results of Data Analysis

### 4.1. Respondents Profile

This study’s respondents are students of public universities pursuing various courses in the northeastern peninsular state of Kelantan, Malaysia. The respondents were from universities such as Universiti Malaysia Kelantan Pengkalan Chepa, University Sains Malaysia Kubang Kerian and University Teknologi MARA, all in the capital city of Kota Bharu, having 9533 student population. As presented here below in Table 2, most of them were females constituting about 59%, with most in the age range of 20 to 30 years (45.3%), next is the age range of 31 to 40 years of age, while 13.4% of them were 41 years and above. With the level of respondents in terms of their study at the university, the majority of them (33.6%) were in their second year of study, with 29.2% Undergraduate 3rd year students followed by those in their first-year stage of study (20.8%) while 13.8% were in the fourth year of their study and about 3% were offering Master/PhD. The respondents’ profile is well represented, with the majority, 44.3% being Malays by race, while 35.9% and 15.1% were Chinese and Indians (refer to Table 2).

### 4.2. Assessment of Measurement Model

The study’s measurement and structural model were analyzed using Smart PLS-SEM version 4. Accordingly, the software can handle non-normality assumptions that are typical of survey research in fields like social science [71]. As suggested by researchers [72,73], we first investigated the likelihood of common method bias by examining the whole interdependence of the study variables in order to address issues of interdependence or collinearity of our single source data. With this approach, each variable undergoes regression on an established variable; thus, if the variance estimation factor value is less than 3.3, there is no bias resulting from using merely one source of data. Given that the computation produced a variance estimation factor (VIF) of below 3.3, single-source bias does not pose a significant problem with the data, as in Table 3 below.

To establish common method variance, a thorough collinearity test was conducted using the PLS-SEM methodology, which was regarded by Kock [73] to be superior to the Harmon single-factor method. Additionally, for data normalcy, web-powered Multivariate Kurtosis or Univariate and Multivariate skewness and kurtosis computation were carried out in accordance with Cain et al. [74]. The results suggested that all the variables were univariate-normal, having their skewness and kurtosis values between 1 and not more than 7 with multivariate normality using the rule of thumb [75] showed a multivariate skewness greater than 3 and multivariate kurtosis also exceeded 20 with Mardia’s multivariate skewness (β = 69.2743, *p* < 0.01) and Mardia’s multivariate kurtosis (β = 126.4454, *p* < 0.01). Hence, the data are therefore judged as being abnormal, which justifies the usage of the PLS bootstrapping method. Subsequently, to validate the reliability of the measurement items and bias behaviour, validity and reliability analysis is performed on the data derived. The details are presented in the following sub-section.

### 4.3. Validity and Reliability

They were further evaluated in terms of validity and reliability of the measurement items because the data showed no evidence of common method bias. The loadings for each of this study variable’s items were checked in accordance with requirements to ensure that items with the least loadings, such as 0.5, were removed in order to reach the necessary VIF [71]. The item PRD-5 from our investigation was eliminated because its loading fell below the allowed loading, as presented here below in Table 4. The variables’ dependability was also investigated [71] using a composite reliability (CR) value that ranged from 0.851 to 0.916, above the minimum value of 0.70. The study also looked at the measuring items’ convergent validity, which is defined as occurring when the average variance extracted (AVE) value is more than 0.50 (refer to Table 4). As demonstrated below in Table 4, with AVE values ranging from 0.68 to 0.745, a good AVE value denotes that, on average, the construct explains more than half of the variance of its indicators. This study was successful in achieving reliability, validity, and convergent validity.

Subsequently, we investigated the discriminant validity of the assessment items in order to confirm the validity and reliability of the study variables. The details are here in the following sub-section.

### 4.4. Discriminant Validity

In order to confirm the validity and reliability of the study variables, we investigated the discriminant validity of the assessment items, this is performed by adopting the Heterotrait–Monotrait (HTMT) criterion. It was determined that the discriminant validity values for the research variables were reasonable, as shown in Table 5 below. Based on [76], the Fornell–Larcker reliability criterion has been criticized; hence, the MTMT matrix approach was used to evaluate discriminant validity using the HTMT ratio of correlations [77]. According to the research, there is a difficulty with discriminant validity when the HTMT value is more than a HTMT0.85 value of 0.85 or a HTMT0.90 value of 0.90 [77,78,79]. Therefore, as indicated in Table 5, all of the values passed the HTMT0.90, and no variables had an HTMT value of 1, indicating the presence of discriminant validity in the constructs [75]. As a result, the row and column values are greater compared with AVE’s both vertically and horizontally, proving the statistical significance of the study’s assessments.

### 4.5. Structural Model of the Tested Hypotheses

The study model was considered adequate after the measuring process, which looked at the constructs’ validity and reliability as well as dealt with problems with data normality. The model produced appropriate findings, which allowed for a bootstrapping approach evaluation of the structural model on the 298 respondents. The tested direct and indirect hypotheses are shown in Table 6 below, along with the study’s structural model, as seen in Figure 1.

### 4.6. Predictive Relevance Using PLS-Predict

Additionally, since PLS version 4 was utilized for the data analysis, a PLS-predict was conducted to determine the predictive relevance of the study model, as illustrated in Table 6 below. We followed the benchmark used in [80] for PLS-predict, which shows that all the independent variables showed Q^2^ > 0 on the two outcome variables (i.e., customer satisfaction-CSAT and Brand loyalty-BRL with the overall Q^2^ for BRL being 0.544 and that of CSAT being 0.746. Again, the RMSE and the PLS-LM values based on the relevant authors all indicate the strong predictive relevance of the study model and also support [80]. Thus, all the errors from the PLS-predict model were lower than the LM model, justifying the strong predictive power of the study model.

### 4.7. Goodness of Fit (GoF) Computation

More so, R Square (R^2^) can also be useful in determining the extent of change of the dependent variable(s) or endogenous variable as a result of the independent variables. The study R-square values on the two endogenous variables accounted by the independent variables for BRL and CSAT from the PLS algorism are 0.622 and 0.755. This signifies that 62.2% and 75.5% of the variance have been explained by the independent variables. According to [71] guidelines for R^2^ values, our study model is termed as substantial. For the study Goodness of fit (GoF), as in Table 7 below, we used the formular GoF = √(AVE × R^2^). With this formular, the AVE value obtained from the measurement model and the average R^2^ values on the endogenous variables are used. This formular has recently been used to calculate GoF as there is no provision for GoF in SmartPLS-SEM-version 4. Based on the computation, the GoF for BRL and CSAT are 0.709 and 0.782 and posits a large GoF of the model following the rule of thumb for interpreting GoF analysis [71,81]. Therefore, we can conclude that our model GoF is described as large, which further supports that our data have a strong predictive relevance and fit the model as expected.

As further demonstrated, we used a 5000-resample PLS bootstrapping approach and followed the recommendations of [79] and other related studies. The tested hypotheses have been presented using path coefficients, standard errors, *t*-values, *p*-values, confidence intervals, F-square (F^2^), and VIF-values for the structural model, which lend support and respond to earlier criticism [80,81]. More importantly, testing study outcomes using more than just the significant *t*-values and *p*-values of the given hypotheses is thought to increase their reliability. Table 8 demonstrates the study’s tested direct hypothesis based on how current research ought to present their hypotheses in light of this. 

The tested hypotheses have been reported based on their path coefficients, the standard errors, *t*-values, *p*-values, confidence intervals, F-square (F^2^) and VIF-values. Table 8 demonstrates that the VIF values for the study’s antecedent variables’ effect on mediating and the dependent variables fall within the maximum threshold of five (5), where [79] reiterated a VIF that exceeds 5 indicates multicollinearity issues of the data. As shown, the ACU and CSAT effect on BRL showed a significant effect with (*ACU-BRL*; *t* = 1.869, *p* = 0.031, F^2^ = 0.022) and ACU-CSAT; *t* = 3.628, *p* < 0.001, F^2^ = 0.130). However, PRD-BRL suggests no significant effect (*t* = 1.452, *p* = 0.073, F^2^ = 0.010), while a significant effect exists between the relationship between PRD-CSAT (*t* = 4.389, *p* < *0*.001 and F^2^ = 0.131). Further, a significant effect exists between INQ-BRL (*t* = 2.485, *p* = 0.006 and F^2^ = 0.030) and the effect of INQ-CSAT (*t* = 1.660, *p* = 0.048, F^2^ = 0.022) also established a significant relationship. Similarly, EPA-BRL registered significant influence (*t* = 2.706, *p* = 0.003, F^2^ = 0.031 but no significant impact of EPA-CSAT (*t* = 0.712, *p* = 0.238 and F^2^ = 0.003). As indicated, an insignificant effect occurred between SPA -> BRL (*t* = 0.966, *p* = 0.167, F^2^ = 0.004) and on the effect of SPA-CSAT (*t* = 1.004, *p* = 0.158, F^2^ = 0.005). Lastly, the effect of CSAT-BRL revealed a significant result (*t* = 6.285 and *p* < *0*.001). Per Cohen’s (1988) effect size (F^2^) guidelines, which give the percentage of alteration clarified by the exogenous variable with F^2^ of 0.02, 0.15 and 0.35, respectively, representing small (0.02), medium (0.15) and large (0.35) effect sizes implies that most of the study variables had small F^2^ on the mediator and the dependent variable. In brief and as shown in Table 8 above, ACU-BRL, ACU-CSAT, PRD-CSAT, INQ-BRL, INQ-CSAT, EPA-BRL, and CSAT-BRL were statistically significant, while PRD-BRL, EPA-CSAT, SPA-BRL, and SPA-CSAT had no significant effect. Subsequently, we tested the indirect or mediating effect of customer satisfaction on the relationship between the antecedent variables as per Table 9 below. 

As presented in above Table 9, the mediating effect of customer satisfaction (CSAT) on the antecedent variables effects along with BRL, the dependent variable are evident. The outcome suggests a significant mediating effect of the first hypothesis; thus, H1b; ACU-CSAT-BRL (β = 0.182, *t* = 2.941, *p* = 0.002). With regard to H2b; PRD-CSAT-BRL (β = 0.174, *t* = 4.021, *p* < 0.001) signifies a significant mediating effect. A positive and significant mediating effect exists with H3b; INQ-CSAT-BRL (β = 0.212, *t* = 3.048, *p* = 0.002). Meanwhile, as shown, an insignificant mediating effect occurred between H4b; EPA-CSAT-BRL (β = 0.028, *t* = 0.695, *p* = 0.243) and H5b; SPA-CSAT-BRL (β = 0.030, *t* = 0.983, *p* = 0.163).

## 5. Discussion

The electronic service quality (E-ServQual) model, also known as service quality (ServQual), was developed by Ismael and Duleba [82] and is used in this study to understand how customers assess the e-service quality of various services, such as the Shopee application. Customer satisfaction (CSAT) and Brand loyalty (BRL) of the Shopee application serve as the study’s mediator and dependent variables, respectively, along with the study’s antecedent variables, the accuracy of delivery order (ACU), price of delivery order (PRD), information quality (INQ), ease of payment (EPA), and security of payment (SPA).

Based on the study outcome, ACU and CSAT significantly influence the BRL of the Shopee app of students of the northern peninsular state of Kelantan, Malaysia context concurs with some past studies on delivery accuracy of orders or the level of timeliness or accuracy of delivery. Hence, it is a symbol to measure the comprehensive strength of any online service provider making the accuracy of delivery a fundamental and integral objective of online customers and significantly influencing customer satisfaction and brand loyalty [44,83].

The insignificant outcome of PRD-BRL is contrary to prior findings [47] that price is one of the factors that can affect a person for the purchase decision and that price significantly influences the purchase decisions and, on the other hand, agreed with the significant effect of PRD-CSAT having a significant effect on the price of delivery and customer satisfaction and brand loyalty. Further, a significant effect exists between INQ-BRL and the effect of INQ-CSAT, which confirms the study conducted by Fauziyah et al. [84]. It establishes a noteworthy impact of customer experience of service quality on brand allegiance and satisfaction. From our research, it is evident that information quality plays a major role in attracting and retaining customer gratification and brand loyalty to the Shopee application as a delivery service to undergraduate students.

Similarly, EPA-BRL is statistically significant (*t* = 2.706, *p* = 0.003, F^2^ = 0.031) but no significant impact of EPA-CSAT (*t* = 0.712, *p* = 0.238 and F^2^ = 0.003). To add on, an insignificant effect occurred between SPA-BRL (*t* = 0.966, *p* = 0.167, F^2^ = 0.004) and the relationship amidst SPA-CSAT (*t* = 1.004, *p* = 0.158, F^2^ = 0.005). The insignificant outcome of security of payment on brand loyalty and customer satisfaction means a lot regarding these respondents’ use of the Shopee application. This supports studies that suggest security is a significant factor that influences customers’ demand for e-commerce, and will purchase or repurchase from an online platform considered to be secure with a good reputation and manages customer information and privacy as expected, for example [41,85]. Lastly, CSAT-BRL revealed a significant result (*t* = 6.285 and *p* < 0.001) and is in tandem with [86], which proved a positive and significant effect of security of payment on customer satisfaction. As expected, for electronic commerce retailers to continue to strive, they need to increase consumer interest in online shopping by improving the quality of their services, providing user-friendly with needed security measures that meet customers’ hopes on time toward customer satisfaction and making them loyal customers of the brand.

The significant mediating effect of CSAT amidst the ACU and BRL (ACU-CSAT-BRL) relationship confirms other studies like [86,87] that a significant effect exists between the price of delivery, customer satisfaction, and brand loyalty. With regard to the significance of PRD-CSAT-BRL, which suggests a significant mediating effect of CSAT, is in line with a previous study [85] on how e-brand experience and in-store experiences influence brand loyalty of novel coffee brands in China along with exploring the roles of customer satisfaction and self–brand congruity and that of on factors that induce customer satisfaction and brand loyalty of the top five express delivery services in China [86].

Similarly, the significant mediating effect of CSAT in H3b thus, INQ-CSAT-BRL agreed with other studies like [88] on the relationship between logistics service quality, customer satisfaction and loyalty in the courier services industry and there there is a significant mediating effect that customer satisfaction has on customers experiences like information quality and e-service quality on brand loyalty [84]. The insignificant mediating effect results of CSAT on H4b; EPA-CSAT-BRL and H5b; SPA-CSAT-BRL are dissimilar with related studies, which found that CSAT positively and significantly mediates the link between EPA and SPA in exploring the effect of brand image relationship on customer loyalty mediated by customer satisfaction [47,48,89]. The findings prove that service quality issues like the accuracy of delivery, delivery costs, payment security and information quality are essential competitive strategies that arouse customer satisfaction and subsequently lead to brand loyalty.

## 6. Contributions and Implications of the Study

This study offers vast contributions or implications which can be categorized as theoretical and practical for Shopee, academicians, and other operators of online service providers as well as SMEs and users of the Shopee app.

### 6.1. Theoretical Implications

This paper seeks to make several theoretical contributions to the literature. For instance, the theoretical contribution of this study for academicians is that this research is expected to provide empirical evidence on the antecedent factors on which consumer satisfaction mediates among online shoppers based on accuracy of delivery, brand loyalty, ease of payment, price of delivery, information quality and security of payment. This novel study focuses on the mediating effect of customer satisfaction, which has received less attention in previous studies, and, especially this study’s use of its antecedent variables. To begin with, our study expanded the application of online service delivery from the theoretical view of online business transactions towards attracting and maintaining loyal customers to their brand. Some previous studies, like [10,11,81], focus on the airline and road transport sectors. Further, most related studies have also focused on the accuracy of delivery orders, prices, and information quality with less incorporation of ease of payment and security of payment with the mediation effect of customer satisfaction and brand loyalty [12,13] with few studies in line with this study on the Shopee applications [1,15], and in Kelantan, the North East state of the Peninsula, Malaysia. On the contrary, prior related studies seldom adopted the TPB and used the technology acceptance model in order to analyze the factors that affect customer satisfaction and brand loyalty [36,43,47]. Our study viewed online customer satisfaction as the main issue based on perceived customer experiences where several customers lodge complaints in relation to Shopee brand loyalty, thereby providing new insights for customers and businesses using the Shopee app.

Likewise, this study theoretically highlighted the essence of ServQual to online service providers as a Shopee application and the need to focus on the accuracy of delivery, price or delivery costs and information quality to attract and satisfy customers and brand loyalty to gain a competitive advantage. Thus, Shopee should work on payment security and ease of payment, making the Shopee Application more reliable than its rivals in delivery services. Lastly, our study’s predictive relevance and Goodness of Fit (GoF) outcome based on the measurement and the structural models constitutes a vital theoretical implication that should prompt future researchers to adopt this study model and the modified measurement items used.

### 6.2. Practical Implications

Practically, this study provides practical implications for Shopee Malaysia and, by extension, online shopping applications in Asia. It is worth paying attention to the quality of application usability in each of its uses, such as speed in responding to customers in product searches and providing convenience in product recommendations based on customers’ purchases and search history by improving customer data privacy policies. A good sense of safety with an electronic black-and-white agreement by Shopee to maintain and not disseminate customer data to other businesses, and the quality and speed of the Shopee app makes payments easier and faster when the application is busy at a certain time. Thus, the practical benefits envisaged from this research include the Shopee app developing sales methods through the website and accepting a companywide information system about customer complaints and swift responses to them to retain loyal customers.

More so, the results suggest that customers are always after safer and easier search engines, secured payment facilities, and quality information on the Shopee applications to provide reliable offline services. These factors can be considered utmost for any service entrepreneur to perform online services and within other sectors. With this, SMEs in logistics can use the Shopee website to establish trust with existing/prospective customers and encourage them to shop. Hence, loyal customers will generate long-term revenues and profits for firms and reduce the cost of operations and acquiring new customers. This study’s findings can be helpful to online delivery businesses in developing or developed countries and enhance the capabilities of their apps by incorporating this model as a strategy to attract loyal customers, such as students, which form the most extensive customer base that buys online.

Our study contributes uniquely to the body of literature on the antecedent factors influencing brand loyalty mediated by customer satisfaction on brand loyalty of online service platforms like the Shopee application among university students, which has little empirical evidence.

Further, our study provides a blueprint for policy and practical implications to online service businesses like Shopee to attract and retain brand-loyal clients, particularly the youth (students), who make up a significant portion of their clientele. Based on the findings, Shopee, like any other online service business, needs not compromise on devising creative ideas to better serve their clients. For practical implications, this innovative study provides Shopee and its affiliated firms with essential information that they can use to attract more customers by putting its main findings into practice. From the relevant literature, Shopee online customers are more interested in quality products; hence, merchants should be unwavering for loyal customers based on service quality, timely delivery, ease of security and payment to lead to brand loyalty backed by fulfilled customers.

## 7. Conclusions

In all, this study has achieved its overall objective by exploring the mediating effect of customer satisfaction on the antecedent factors of Shopee brand loyalty. The outcomes of this paper support earlier studies and provide new theoretical and practical implications for Shopee online express and its related customers and business partners and future research on the role of online client satisfaction and brand loyalty in the services sector of businesses.

## 8. Limitations and Recommendations

The aims of this study have been achieved in relation to the mediating role of customer satisfaction on antecedent factors of brand loyalty among Shopee customers. However, as phenomenal of every research, though the empirical findings of this study contribute to the existing pull of literature, a few limitations and recommendations can be discovered. To begin with, as the study respondents were university students from the northeastern Peninsula of Malaysia, its findings cannot be extrapolated. We recommend that for continuous customer satisfaction and referrals to the Shopee app for brand loyalty, Shopee should be more concerned about customer needs, provide appropriate and timely information to customer complaints while ensuring that their products and services are of desired quality. Shopee should not compromise customer data privacy policy with a guaranteed security agreement endorsed and strictly adhered to by Shopee and customers that Shopee, under no circumstances, will divulge their data and will maintain and not disseminate customer data to their disadvantage. Again, future researchers can use our unique research framework for different states or Malaysia and in other countries and with other online service apps like Mudah.com, Tiktok, and Lazada, among others, to generalize these research findings. Longitudinal or mixed methods with the Shopee firm and allied service delivery and logistic SMEs study may also give further insight into our findings with other mediating, moderating and control variables in using this study model.

## Figures and Tables

**Figure 1 behavsci-13-00563-f001:**
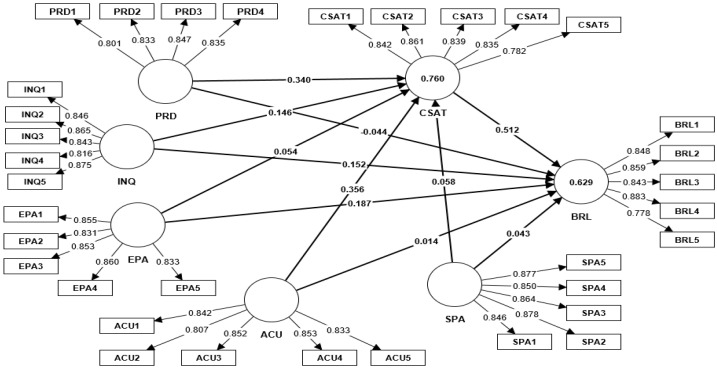
Study Structural Model.

**Table 1 behavsci-13-00563-t001:** Pre-test results for reliability and validity of variables (*n* = 40).

Variable Name	Items per Variable Used	Cronbach Alpha Value (α) per Variable
Accuracy of Delivery (ACU)	5	0.829
Brand Loyalty (BRL)	5	0.908
Customer Satisfaction (CSAT)	5	0.844
Ease of payment (EPA)	5	0.727
Information Quality (INQ)	5	0.814
Price of Delivery (PRD)	5	0.794
Security of Payment (SPA)	5	0.834

**Table 2 behavsci-13-00563-t002:** Respondent profiles (*n* = 298).

Item	Occurrence	Percent	Item	Occurrence	Percent
Gender Female Male	176 122	59.1 40.9	Age Range Less than 20 20–30 31–40 41 and above	18 135 105 40	6.0 45.3 35.2 13.4
Stage of Study 1st year Undergraduate 2nd year Undergraduate 3rd year Undergraduate 4th year Undergraduate Master/PhD	62 100 87 41 8	20.8 33.6 29.2 13.8 02.7	Race Malay Chinese Indian Others	132 107 45 14	44.3 35.9 15.1 4.7

**Table 3 behavsci-13-00563-t003:** Testing for full collinearity.

ACU	BRL	CSAT	EPA	PRD	INQ	SPA
3.405	2.145	3.581	1.295	1.988	1.629	1.998

Note: ACU: accuracy of delivery; BRL: brand loyalty; CSAT: customer satisfaction; EPA: ease of payment; PRD/FRD: price of delivery; INQ: information quality, SPA: payment security.

**Table 4 behavsci-13-00563-t004:** Reliabilities of study variables.

Construct	Items Used (IU)	Loadings (LDS)	Composite Reliability (CR)	Average Variance Extracted (AVE)
Accuracy of Delivery (ACU)	ACU1 ACU2 ACU3 ACU4 ACU5	0.842 0.807 0.852 0.853 0.833	0.894	0.701
Brand Loyalty (BRL)	BRL1 BRL2 BRL3 BRL4 BRL5	0.848 0.859 0.843 0.883 0.778	0.898	0.711
Customer Satisfaction (CSAT)	CSAT1 CSAT2 CSAT3 CSAT4 CSAT5	0.842 0.861 0.839 0.835 0.782	0.889	0.692
Ease of payment (EPA)	EPA1 EPA2 EPA3 EPA4 EPA5	0.855 0.831 0.853 0.860 0.833	0.902	0.717
Price of Delivery (PRD)	PRD1 PRD2 PRD3 PRD4	0.801 0.833 0.847 0.835	0.851	0.687
Information Quality (INQ)	INQ1 INQ2 INQ3 INQ4 INQ5	0.846 0.865 0.843 0.816 0.875	0.904	0.721
Security of Payment (SPA)	SPA1 SPA2 SPA3 SPA4 SPA5	0.846 0.878 0.864 0.850 0.877	0.916	0.745

Note: Item PRD5 was deleted as it did not meet the item loading requirement.

**Table 5 behavsci-13-00563-t005:** Discriminant validity using HTMT ratio.

Used Variables	1	2	3	4	5	6	7
1. Accuracy of delivery (ACU)							
2. Brand Loyalty (BRL)	0.853						
3. Customer satisfaction (CSAT)	0.827	0.848					
4. Ease of payment (EPA)	0.790	0.755	0.792				
5. Price of delivery (PRD)	0.711	0.750	0.754	0.790			
6. Information quality (INQ)	0.702	0.721	0.732	0.748	0.832		
7. Security of payment (SPA)	0.771	0.691	0.756	0.621	0.759	0.837	

**Table 6 behavsci-13-00563-t006:** PLS predict for predictive relevance.

Construct	Q^2^_Predict	RMSE		
BRL	0.544	0.683		
CSAT	0.746	0.511		
**ITEM**	**RMSE**	**RSME**	**PLS-LM**	**Q^2^_predict**
BRL1	1.245	1.283	−0.038	0.387
BRL2	1.163	1.214	−0.051	0.372
BRL3	1.251	1.279	−0.028	0.336
BRL4	1.225	1.285	−0.060	0.399
BRL5	1.234	1.245	−0.011	0.414
CSAT1	0.704	0.736	−0.032	0.491
CSAT2	0.642	0.688	−0.046	0.542
CSAT3	0.749	0.792	−0.043	0.467
CSAT4	0.681	0.709	−0.028	0.534
CSAT5	0.676	0.682	−0.006	0.518

**Table 7 behavsci-13-00563-t007:** Goodness of Fit (GoF) Computation Table.

Variables Used	Each AVE Value	R-Square (R^2^)
1. Accuracy of delivery (ACU)	0.701	
2. Brand Loyalty (BRL)	0.711	0.622
3. Customer satisfaction (CSAT)	0.692	0.755
4. Ease of payment (EPA)	0.717	
5. Price of delivery (PRD)	0.687	
6. Information quality (INQ)	0.721	
7. Security of payment (SPA)	0.745	
AVE Average Scores	0.711	
AVE × R^2^ (BRL)	0.442	
AVE × R^2^ (CSAT)	0.537	
GoF Calculation = √(AVE × R^2^) BRL	0.665	
GoF Calculation = √(AVE × R^2^) CSAT	0.733	

**Table 8 behavsci-13-00563-t008:** Direct effect of tested relationships.

Hypotheses	Std Beta (β)	Std Dvn	*t*-Value	*p*-Value	Bcill	Bcull	F2	VIF	Decision
Tested									
1a1. ACU-BRL	0.196	0.105	1.869	0.031	0.023	0.367	0.022	4.043	Yes, Ok
1a2. ACU-CSAT	0.356	0.098	3.628	*p* < 0.001	0.198	0.52	0.13	4.043	Yes, Ok
2a1. PRD-BRL	0.13	0.089	1.452	0.073	−0.021	0.271	0.010	3.729	No
2a2. PRD-CSAT	0.34	0.077	4.389	*p* < 0.001	0.214	0.465	0.131	3.729	Yes, Ok
3a1. INQ-BRL	0.226	0.091	2.485	0.006	0.073	0.368	0.030	3.962	Yes, Ok
3a2. INQ-CSAT	0.146	0.088	1.66	0.048	0.002	0.285	0.022	3.962	Yes, Ok
4a1. EPA-BRL	0.214	0.079	2.706	0.003	0.1	0.362	0.031	3.445	Yes, Ok
4a2. EPA-CSAT	0.054	0.076	0.712	0.238	−0.063	0.186	0.003	3.445	No
5a1. SPA-BRL	0.073	0.075	0.966	0.167	−0.06	0.19	0.004	2.844	No
5a2. SPA-CSAT	0.058	0.058	1.004	0.158	−0.041	0.147	0.005	2.844	No
6. CSAT-BRL	0.512	0.081	6.285	*p* < 0.001	0.369	0.639			Yes, Ok

Note: Bcill = bias-corrected lower level and Bcull = bias-corrected upper level.

**Table 9 behavsci-13-00563-t009:** Indirect effect of tested relationships.

Hypo	Relationships	Stdt β	Std Dvn	*t*-Values	*p*-Values
H1b	ACU-CSAT-BRL	0.182	0.062	2.941	0.002
H2b	PRD-CSAT-BRL	0.174	0.043	4.021	*p* < 0.001
H3b	INQ-CSAT-BRL	0.212	0.07	3.048	0.002
H4b	EPA-CSAT-BRL	0.028	0.04	0.695	0.243
H5b	SPA-CSAT-BRL	0.030	0.03	0.983	0.163

## Data Availability

The data of this study are available from the first author upon reasonable request.

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
