# Peer review of "The Mediating Role of Customer Satisfaction between Antecedent Factors and Brand Loyalty for the Shopee Application"

_behavsci, 2023, doi:10.3390/bs13070563_

Round 1
Reviewer 1 Report
The dynamic and continuous development of new technologies explains the timeliness and validity of the research issue undertaken. It is still necessary to conduct analyses from which recommendations for mobile application developers are derived. Taking into account in recommendations the expectations of users of applications undoubtedly increases their usability and builds customer satisfaction. I consider it important to conduct analysis on factors that affect brand loyalty in online shopping applications. This is a requirement of modern times because interest in mobile applications is growing. This tool needs to be improved in relation to the expectations of users and therefore research should be conducted.
The analysis of the literature on the subject was done correctly, the research hypotheses and their verification and the model were correctly formulated.
However, I did not find a directly defined purpose of this study. One can, of course, infer what the purpose is, but it is better to write about it.
I also suggest separating after the discussion and conclusions a chapter : Implications and limitations. This content is included in the last chapter, but it would be advisable to extract it.
Author Response
Attached the Response to the Reviewer 1

Reviewer 2 Report
Dear Authors,
Thank you for providing this paper on the "mediating role of customer satisfaction between antecedent factors and brand loyalty of shopee application". The topic of this paper is interesting and up-to date. However, the following points are worthy of consideration to improve the manuscript's comprehensibility.
[1]. The literature should be enhanced by emphasizing recent studies, which have examined the determinants of brand loyalty. Examples of papers are given below:
· https://doi.org/10.1016/j.cstp.2022.05.006
· https://doi.org/10.1177/0972150919892689
[2]. From line 153 to line 156, "Privacy or security refers, […], for online banking customers", there is no in-text citation to support the claims. Could provide supporting references for this paragraph.
[3]. Since data were collected using Google Forms, I encourage authors to provide more details about the questionnaire pre-testing.
[4]. For more clarity, I encourage authors to include the original figure 1
[5]. In the Data Analysis section, the authors are encouraged to introduce other essential criteria for checking the structural model, including the coefficient of determination (R2) , the predictive relevance (Q2), and the model goodness-of-fit (GoF).
[6]. The current version of the conclusion and discussions require further development. As such, I encourage the authors to consider the study implications from a theoretical and practical point of view.
All the best.
Minor editing of English language required.
Author Response
Attached the Response to the Reviewer 2

Reviewer 3 Report
This manuscript has aimed to investigated the antecedents of brand loyalty of Shopee appplication, the research topic is interesting, the literature is thoroughly reviewed, the hypotheses are relatively reasonably developed, the methodology is acceptable, the data analysis is proper and clearly reported, and the conclusions are well supported by the results. Overall, this manuscript is acceptable, but there are still several issues to be addressed.
1. There are alreadly quite a few studies on the antecedents of brand loyalty of shoping application, and the mediating effect of customer satisfaction is not novel enough, therefore, the originality of this study is not very high. So, the authors should discuss more about the relationship and differences of this study and the previous studies, and highlight the theoretical contribution of this study.
2. Putting H1b right after H1a is not a good idea, in fact, the single paragraph of discussion of the effect of ACU on BRL is not conving enough, and seldomly talk about the mediating effect CSAT. The same problem occors in H2b, H3b, H4b, H5b. It will be better to put these hypotheses after H6.
3. The research objects(students) was highlighted in the hypotheses, what is the theoretical differences between students and other groups of people? It seems that other people also values the Accuracy of Delivery Order, Price of Delivery, Information Quality, Ease of Payment, and Payment Security. If you can not identify some unique variables for a specific group, it may be not necessary to highligh it.
4. The details of the development of measurement scales should be reported, at least the final items should be reported in an appendix.
5. Figure 1 should be redraw with a professional solfware.
6. More discussion about the practical implication, limitations, future directions shoud be added.
Author Response
Attached the Response to the Reviewer 3

Round 2
Reviewer 2 Report
The authors have produced a thorough response that significantly enhances the paper. I am now happy to accept the paper in this form.
Minor editing of English language required.